# Laser Cavitation Peening and Its Application for Improving the Fatigue Strength of Welded Parts

**Hitoshi Soyama** 

Department of Finemechanics, Tohoku University, Sendai 980-8579, Japan; soyama@mm.mech.tohoku.ac.jp;
Tel.: +81-22-795-6891; Fax: +81-22-795-3758

**Abstract:** During conventional submerged laser peening, the impact force induced by laser ablation is used to produce local plastic deformation pits to enhance metallic material properties, such as fatigue performance. However, a bubble, which behaves like a cavitation, is generated after laser ablation, known as "laser cavitation." On the contrary, in conventional cavitation peening, cavitation is generated by injecting a high-speed water jet into the water, and the impacts of cavitation collapses are utilized for mechanical surface treatment. In the present paper, a mechanical surface treatment mechanism using laser cavitation impact, i.e., "laser cavitation peening", was investigated, and an improvement in fatigue strength from laser cavitation peening was demonstrated. The impact forces induced by laser ablation and laser cavitation collapse were evaluated with a polyvinylidene fluoride (PVDF) sensor and a submerged shockwave sensor, and the diameter of the laser cavitation was measured by observing a high-speed video taken with a camera. It was revealed that the impact of laser cavitation collapse was larger than that of laser ablation, and the peening effect was closely related to the volume of laser cavitation. Laser cavitation peening improved the fatigue strength of stainless-steel welds.

**Keywords:** surface modification; laser peening; cavitation peening; welding; fatigue; stainless steel

## 1. Introduction

The weld points between metallic material components are the weakest in terms of fatigue properties [1], as welding causes residual tensile stress [2–4]. It has been reported that the fatigue strength of the heat-affected zone (HAZ) is higher than that of base metal in some cases; however, the effect not only occurs at the microstructure of the HAZ, but causes welding defect and stress concentration [5]. Mechanical surface treatment such as peening can improve the fatigue properties by introducing compressive residual stress and welding line deformation. Thus, it is worthwhile developing a new peening technique, such as laser peening, which can control peening area, to enhance the fatigue life and strength of welds, as the peening area required is limited to near the HAZ region.

Ultrasonic impact peening [6,7], hammer peening, and burnishing [8] are applicable for the post-processing of welds to reduce tensile residual stress and notch effects in welds. Water jet peening also reduces the residual stress of 304 stainless steel welds [9]. Improvement in stainless steel weld fatigue strength using a water jet peening method has been reported [10]. The fatigue life of 316L stainless steel welds treated by water jet peening using a 300 MPa water jet, cavitation peening using a 30 MPa submerged jet, and shot peening using shots accelerated by an 8 MPa water jet was compared in a previous paper [10]. Water jet peening refers to a peening method using a water column's impingement impact using a water jet, and cavitation peening refers to a peening method using the cavitation bubble collapse impact [11,12]. Note that cavitation is still a big problem for hydraulic machinery, as cavitation impacts produce erosion [13]. However, cavitation impacts can be utilized for peening in the same way as shot peening. The great advantage of cavitation peening is lower surface roughness, as no shots are used [14].

During conventional cavitation peening, a submerged high-speed water jet with cavitation, i.e., a cavitating jet [15,16], has been used to generate cavitation, a kind of "hydrodynamic cavitation." Soyama created a cavitating jet in air by injecting a high-speed water jet into a low-speed water jet injected into air, using a concentric nozzle [17,18]; research on the cavitating jet in air was followed by Marcon et al. [19,20]. It has been reported that the cavitating jet in air improves the fatigue properties of friction stir welded aluminum alloy [21]. Information about peening using ultrasonic cavitation can be found in the review by Soyama [12].

There are three types of laser peening; peening methods using a pulse laser: water film types [22–25], in which a target is covered with a water film, a pulse laser is irradiated, and a shockwave induced by laser ablation is trapped by the water's inertial force, causing plastic deformation of the target; submerged laser peening [26,27], in which the target is placed in water, which has been applied to mitigate stress corrosion cracking in nuclear power plants [28]; a laser peening method without a sacrificial overlay under atmospheric conditions using a femtosecond laser [29]. Submerged laser peening produces a bubble, which behaves like a cavitation bubble, after laser ablation and produces a bubble collapse impact [11,30]. In the present paper, the bubble caused by the pulse laser is called "laser cavitation." In the case of submerged laser peening, it is believed that the peening process occurs due to laser ablation [27], as the amplitude of the pressure wave induced by laser ablation is bigger than that of laser cavitation, when the pressure wave in water is measured by a submerged shockwave sensor [30]. However, the impact generated by laser cavitation is larger than that of laser ablation [11], when the impact through in the target is measured by a polyvinylidene fluoride (PVDF) sensor, originally developed to evaluate cavitation impacts [31,32]. Namely, in the case of submerged laser peening, one pulse laser can hit twice, i.e., laser ablation and laser cavitation. When conditions are optimized, the impact of laser cavitation is larger than that of laser ablation [11]. As the impact force direction of cavitation can be controlled by the boundary condition [33], the impact of laser cavitation at submerged laser peening can be removed by setting a boundary wall around laser cavitation. When the peening intensity of the submerged laser peening was measured in a previous study, the peening intensity of the submerged laser peening without laser cavitation impact was nearly half of that with laser cavitation impact [34].

On the contrary, submerged laser peening can use laser cavitation impact without laser ablation by focusing water near the target [35]. It has been reported that laser cavitation without laser ablation introduces compressive residual stress into austenitic stainless steel SUS316L [36]. Thus, optimized submerged laser peening is a kind of cavitation peening that uses laser cavitation. In the present paper, cavitation peening using laser cavitation is referred to as "laser cavitation peening".

During conventional submerged laser peening, the second harmonic wavelength of a Q-switched Nd:YAG laser, i.e., 532 nm, is used to minimize laser absorption by water. An Nd:YAG laser's fundamental harmonic wavelength, i.e., 1064 nm, has been used to investigate bubble behavior during cavitation generation [37], as the heat is more concentrated. Note that 40% of the pulse energy at 1064 nm is lost to obtain 532 nm during wavelength conversion. When using the impact of laser cavitation, it should be considered which wavelength is better, 532 nm or 1064 nm, by considering the absorption of the laser by water and the energy loss due to wavelength conversion.

This paper consists of two parts. The first demonstrates the importance of laser cavitation during submerged laser peening, i.e., laser cavitation peening, considering the relation between the size of the laser cavitation and the peening intensity using wavelengths of 532 and 1064 nm of the Nd:YAG laser. The second demonstrates the improvement of weld fatigue properties by laser cavitation peening.

## 2. Laser Cavitation Peening

### 2.1. Experimental Apparatus and Procedures to Compare Laser Ablation and Laser Cavitation

Figure 1 illustrates a schematic diagram of the submerged laser peening system used to investigate peening intensity. The used laser pulse source (Continuum, San Jose, CA, USA) was a Q-switched Nd:YAG laser with wavelength conversion. By controlling the wavelength conversion, both the 532 nm and 1064 nm wavelengths could be obtained. The pulse width, the beam diameter, and the laser pulse's repetition frequency were 6 ns, 6 mm, and 10 Hz, respectively. The maximum energy was 0.2 J at 532 nm and 0.35 J at 1064 nm. Using reflective mirrors, the pulse laser was expanded by a concave lens and focused onto the specimen placed in a water-filled chamber made of 3 mm thick hard glass using convex lenses to avoid damage to the glass of the chamber. The focal length of the final convex lens was 100 mm. The distance in air ($s_a$) from the final convex lens to the glass and the distance in water ($s_w$) from the glass to the target were optimized to enhance the peening intensity. A handmade PVDF sensor [31,32] was installed between the target and the holder to measure the impact passing through the specimen. The specimen was tightened to the holder with the PVDF sensor by screws. The PVDF sensor was not calibrated to reveal the impact between laser ablation and laser cavitation collapse. A submerged shockwave sensor (Dr. Müller Instruments GmbH & Co. KG, Oberursel, Germany) was also placed in the chamber to evaluate the pressure wave propagation in the water. A high-speed video camera observed the laser ablation and laser cavitation aspects induced by the pulse laser at 230,000 frames per second. The laser cavitation's maximum diameter was evaluated by measuring the bubble size's image recorded by the high-speed video camera. The developing time of laser cavitation ($t_D$) was measured using the sensor's signal recorded by a digital oscilloscope (Tektronix, Inc., Beaverton, OR, USA).

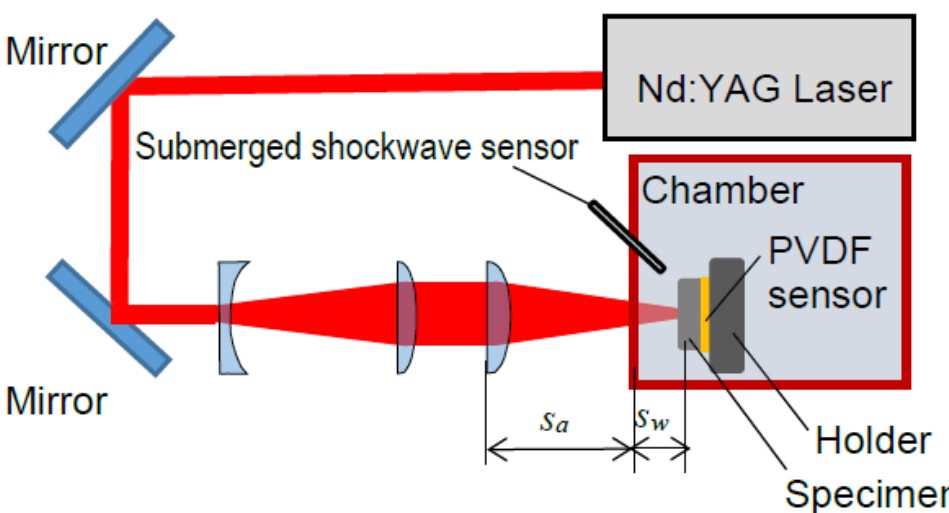

**Figure 1.** Schematic diagram of the submerged laser peening system. PVDF, polyvinylidene fluoride; $s_a$, distance in air; $s_w$, distance in water.

An Almen gage of N strip [38] and a plate specimen made of Japanese Industrial Standards JIS SUS316L stainless steel and A2017-T3 duralumin were treated and had their arc height measured by a test strip gage [38], and were then used to evaluate peening intensity by laser cavitation peening. The size of the tested materials was $50 \times 20 \times 3$ mm for stainless steel, $50 \times 20 \times 5$ mm for duralumin, and $76 \times 19 \times 0.8$ mm for the N strip. At the impact measurement, one or two specimens with changing laser powers were used for each material with a moving specimen so that the pulses did not overlap. At the arc height measurement, six to seven specimens were used for each material with changing laser power. The N strip or the plate specimen was placed on a stage with stepping motors and moved orthogonally with respect to the laser beam. The pulse density ($d_L$) was

calculated from the speed ($v_s$) and stepwise distance ($d_S$) of the stepping motor, and the repetition frequency was 10 Hz, as seen in Equation (1).

$$d_L = \frac{10\,n}{v_s \cdot d_S} \tag{1}$$

where $n$ is the number of scans.

### 2.2. Results and Discussions Comparing with Laser Ablation and Laser Cavitation

Figures 2 and 3 show laser ablation (LA) and laser cavitation (LC) produced by pulse lasers at λ = 532 nm and 1064 nm. Each figure shows (a) LA and LC changing with time ($t$), (b) signal from the PVDF sensor, and (c) signal from the submerged shockwave sensor. At both λ = 532 and 1064 nm, LC developed after LA and reached maximum size at $t$ = 0.48 ms for λ = 532 nm and $t$ = 0.50 ms for λ =1064 nm, then it collapsed at $t$ = 0.96 ms for λ = 532 nm and $t$ = 1.00 ms for λ = 1064 nm, respectively. At both λ = 532 and 1064 nm, the amplitude of the PVDF sensor signal and the submerged shockwave sensor had peaks at LA and the first LC collapse. As shown in Figures 2b and 3b, the PVDF sensor detected the second LC collapse at $t$ = 1.49 ms for λ = 532 nm and $t$ = 1.52 ms for λ = 1064 nm, respectively, although the submerged shockwave sensor could not detect the second collapse of LC. It can therefore be concluded that the PVDF sensor could evaluate the impact more accurately than the submerged shockwave sensor.

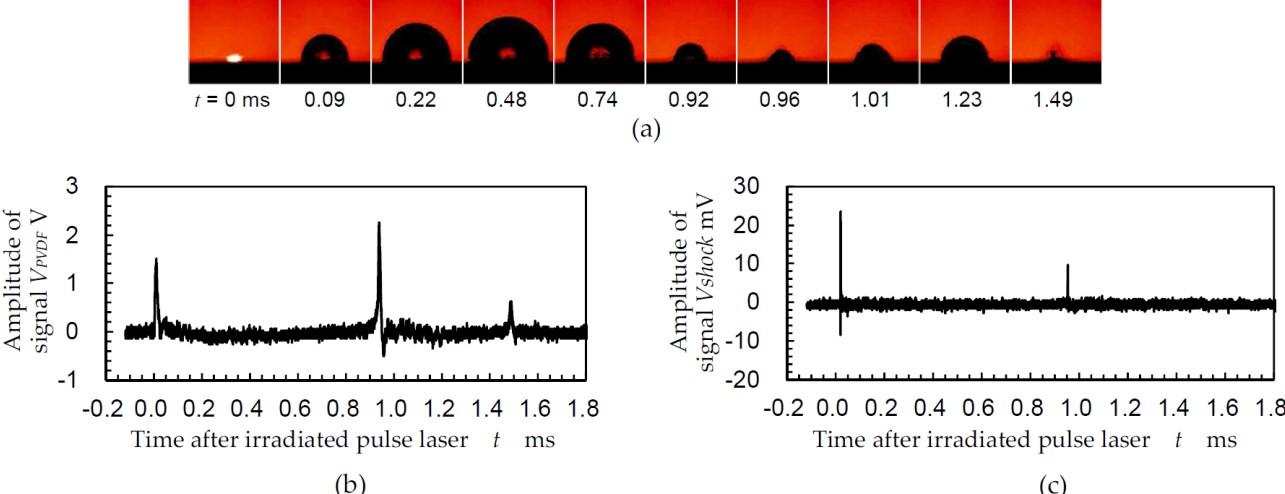

**Figure 2.** Laser ablation (LA) and laser cavitation (LC) produced by pulse laser of λ = 532 nm. (**a**) LA and LC observed through neutral density filter; (**b**) signal from a polyvinylidene fluoride (PVDF) sensor; (**c**) signal from submerged shockwave sensor.

Note that the noise levels of λ = 532 and 1064 nm were slightly different, as the specimen and PVDF sensor were tightened to the holder by screws. The other important result was that the amplitude of the signal from the PVDF sensor at the first LC collapse was larger than that of LA, although the amplitude of the signal from the submerged shockwave sensor at LA was larger than that of the first LC collapse, in agreement with [30]. As mentioned above, as the PVDF sensor could detect the impact more accurately than the submerged shockwave sensor, the first LC collapse was larger than that of LA. Namely, with optimized submerged laser peening, the impact induced by the first LC collapse was more effective than that of LA. The vibration induced by the bubble collapse did not reduce the peening intensity, as the vibration frequency induced by the bubble collapse was much higher than the repetition of the laser.

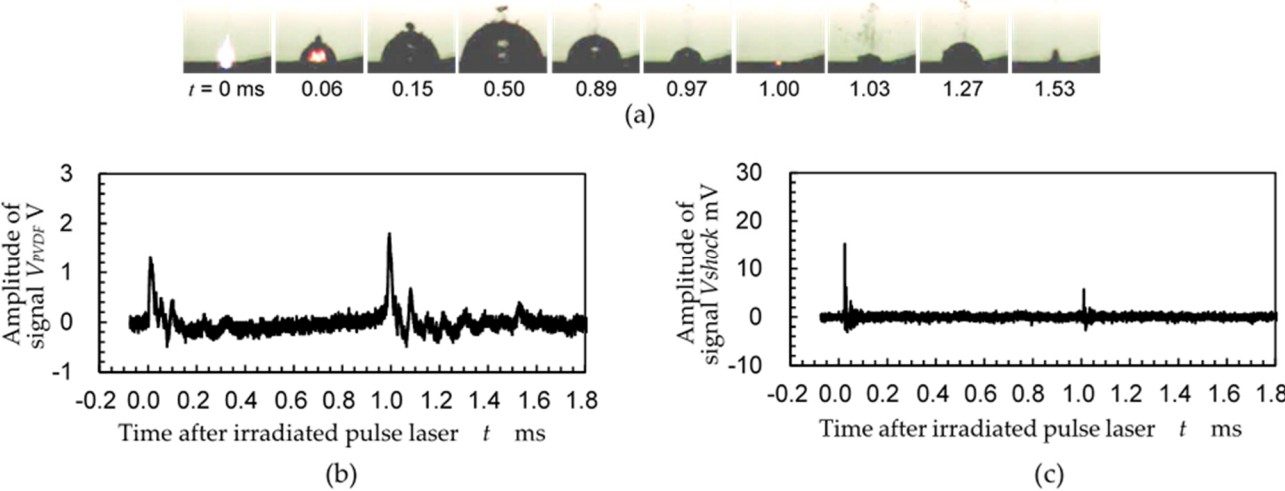

**Figure 3.** LA and LC produced by a pulse laser at λ = 1064 nm. (**a**) LA and LC; (**b**) signal from a PVDF sensor; (**c**) signal from submerged shockwave sensor.

Revealing a key parameter of LC, Figure 4 illustrates the relationship between the developing time of LC ($t_D$) (μs), and the maximum diameter of LC ($d_{max}$) (mm) induced by the pulse laser. As the bubble was developed and then shrunk, $d_{max}$ was defined by the diameter which had a maximum with the time. In Figure 4, "hemispherical bubble" refers to the hemispherical bubble induced by the pulse laser that irradiated the target and developed, as shown in Figures 2a and 3a. The other symbols show the spherical bubbles produced by the pulse laser, which was focused in water without the target changing with water depth. Ion-exchanged water was used. The air content ratio and the temperature of the test water were controlled by degassing membrane and a chiller to examine the water quality. All symbols were almost on the line, as calculated by Equation (2).

$$d_{max} = 0.0103\, t_D \tag{2}$$

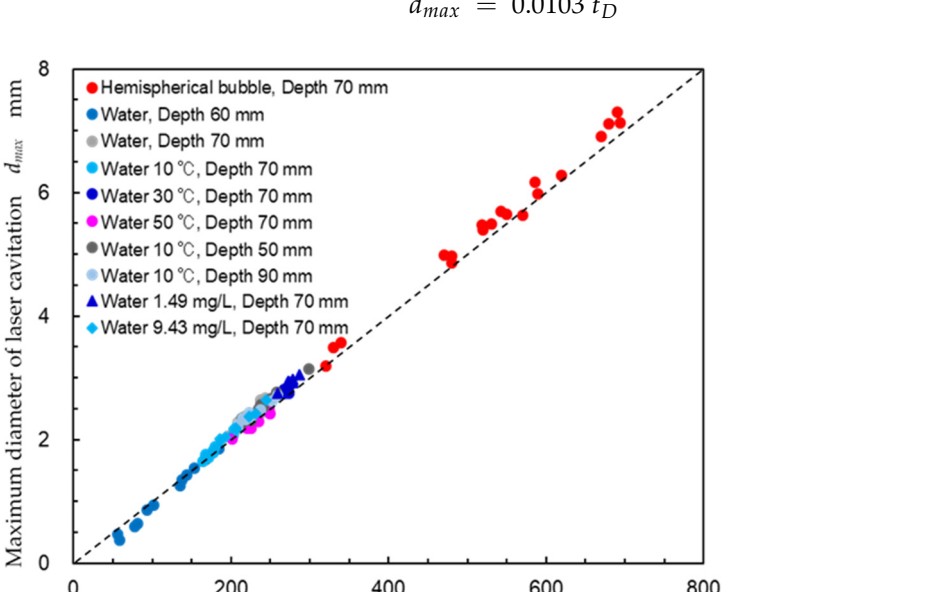

**Figure 4.** Relationship between the developing time $t_D$ and the maximum diameter $d_{max}$ of LC.

It could be concluded that the maximum diameter of LC was proportional to the LC development time ($t_D$), which was defined by the time from laser ablation to the first LC collapse. Thus, in the present paper, $t_D$ was used as the main parameter of LC.

In order to compare capability to generate the laser cavitation of the pulse laser using $\lambda = 532$ nm and 1064 nm, Figure 5 shows $t_D$ as a function of the relative energy of the source laser ($E_L$); the $t_D$ of $\lambda = 1064$ nm was larger than that of $\lambda = 532$ nm. Namely, the pulse laser using $\lambda = 1064$ nm could generate a larger LC, as $d_{max}$ can be calculated by $t_D$. As larger cavitations generate larger impacts [39–42], a wavelength of 1064 nm should be better than 532 nm from an LC generation perspective. Thus, $\lambda = 1064$ nm was chosen for the following experiment. At $E_L > 0.8$, $t_D$ of $\lambda = 1064$ nm was slightly lower than that of $\lambda = 532$ nm, as the small bubble was generated near the focal point due to heat convergence, and it could be reduced by optimizing the focal length of the final convex lens.

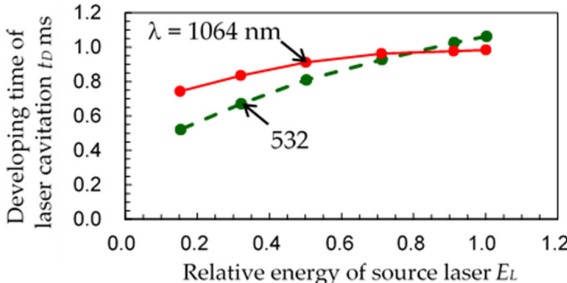

**Figure 5.** Developing time of laser cavitation as a function of the relative energy of the source laser.

The effects of the signals from the PVDF sensor and the submerged shockwave sensor on peening intensity were investigated. Figure 6 illustrates the relationship between arc height $h$ and amplitude of the signal from (a) the PVDF sensor and (b) the submerged shockwave sensor. Table 1 shows the correlation coefficient between the arc height and output signal from sensors. The specimens were treated with 4 pulses/mm², and the tape of the N strip refers to black aluminum tape with a thickness of 80 μm, as black aluminum tape has been used for conventional laser peening with a water film to protect the surface. As shown in Figure 6a, both the LA and LC signals from the PVDF sensor were proportional to the arc height. As shown in Table 1, the correlation coefficient of the signal from the PVDF sensor was larger than that of the submerged shockwave sensor, except for the N strip with tape, and the averaged value of the PVDF sensor was larger than that of the submerged shockwave sensor. It can be concluded that the signal from the PVDF sensor reveals the peening effect.

**Table 1.** Correlation coefficient between the arc height and output signal from the sensors.

| Materials | PVDF Sensor | | Submerge Shockwave Sensor | |
|:---:|:---:|:---:|:---:|:---:|
| | LA | LC | LA | LC |
| Duralumin | 0.961 | 0.982 | 0.913 | 0.299 |
| Stainless steel | 0.925 | 0.947 | 0.881 | 0.381 |
| N strip with tape | 0.927 | 0.892 | 0.933 | 0.130 |
| N strip without tape | 0.983 | 0.963 | 0.937 | 0.146 |
| Averaged value | $0.95 \pm 0.03$ | $0.95 \pm 0.04$ | $0.92 \pm 0.03$ | $0.24 \pm 0.12$ |

In order to investigate the effect of LC on the peening intensity, Figure 7a shows the relationship between the developing time of LC ($t_D$) and the arc height ($h$). As calculated by Equation (2), the $t_D$ was proportional to the diameter of the LC. The specimens were treated with 4 pulses/mm². When the following power law was assumed as the relationship between $t_D$ and $h$ as shown in Equation (3), the exponent $n$ was 3.3 for duralumin, 4.1 for stainless steel, 6.4 for the N strip without tape, and 7.0 for the N strip with tape.

$$h \; \propto \; t_D{}^n \tag{3}$$

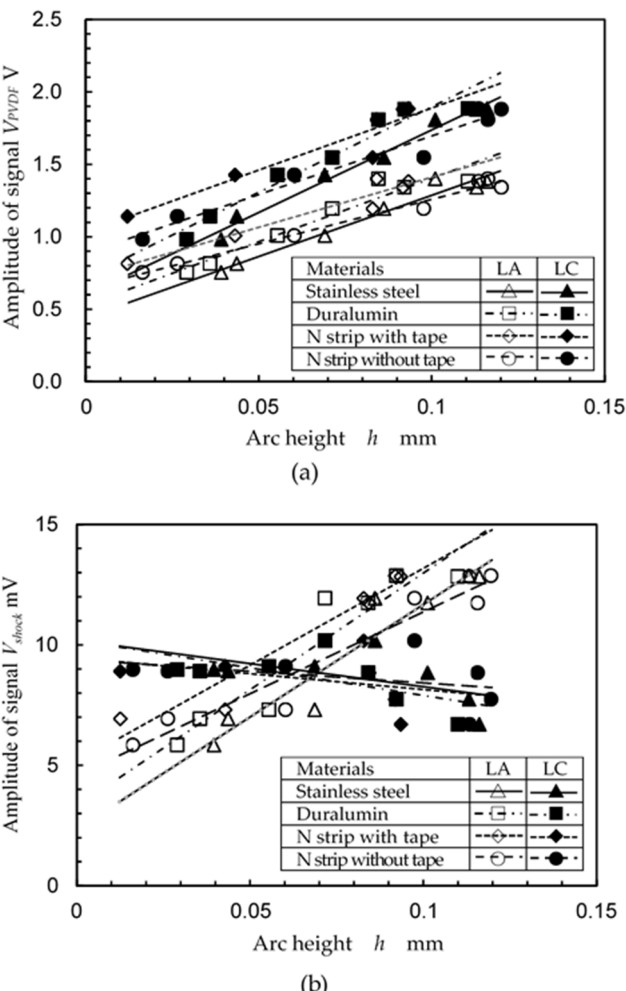

**Figure 6.** Relationship between the arc height and amplitude of the output signal from sensors. (**a**) PVDF sensor; (**b**) submerged shockwave sensor.

It has been reported that the aggressive intensity of cavitation is proportional to the volume of cavitation [15,40,42] and the threshold level of the erosion caused by cavitation impact [43]. Therefore, Equation (4) was assumed.

$$h \propto (t_D - t_{th})^3 \qquad (4)$$

where $t_{th}$ was the threshold level, obtained using the least square method (Figure 7a). Equation (4) demonstrates that weak cavitation with a diameter smaller than 0.0103 $t_{th}$ in mm, considering Equation (2), would scarcely deform a material. The $t_{th}$ is shown in Table 2 and Figure 7b (dashed-dotted line in the same color as the symbol). There was a correlation between surface Vickers hardness without peening and threshold level (Figure 7a,b); the correlation coefficients are shown in Table 2.

The threshold diameter $d_{th}$ (mm) was calculated from $t_{th}$ (μs) by the relationship between the developing time of LC and the diameter of LC to investigate the threshold level, as described in Equation (2), and the threshold levels in $t_{th}$ μs and $d_{th}$ mm are shown in Table 2. The correlation was improved by considering the threshold level (Figure 7b). As the number used in the dataset was six or seven in the present study, the probability of a non-correlation was less than 0.005% for duralumin, 0.046% for stainless steel, 0.003% for the N strip with tape, and 0.141% for the N strip without tape. As the probability of a non-correlation was less than 1%, it can be concluded that the relationship was highly

significant. Thus, it can be concluded that the relationship between the developing time of LC considering the threshold level and the arc height was highly significant.

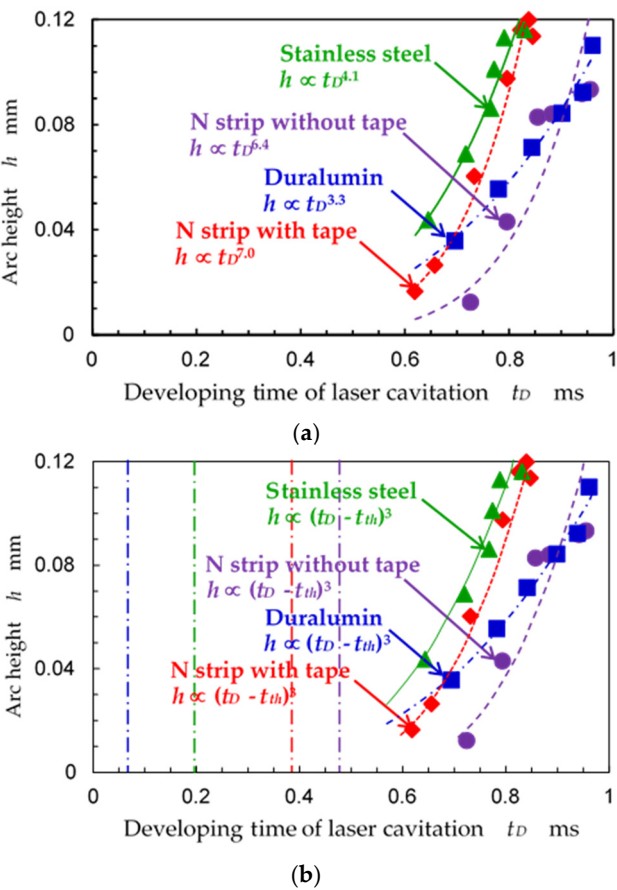

**Figure 7.** Relationship between the developing time of laser cavitation and the arc height. (**a**) The power law without considering the threshold level; (**b**) the power law considering the threshold level.

**Table 2.** Relationship between surface Vickers hardness *Hv* and threshold level.

| Materials | Correlation Coefficient without Considering Threshold Level | Correlation Coefficient Considering Threshold Level | Probability of Non-Correlation of Considering Threshold Level | Surface Vickers Hardness *Hv* | Threshold Level | |
|---|---|---|---|---|---|---|
| | | | | | $t_{th}$ μs | $d_{th}$ mm |
| Duralumin | 0.995 | 0.995 | 0.005% | $129.4 \pm 3.0$ | 64 | 0.66 |
| Stainless steel | 0.982 | 0.983 | 0.046% | $175.2 \pm 1.3$ | 198 | 2.04 |
| N strip with tape | 0.992 | 0.996 | 0.003% | $424.4 \pm 4.7$ | 384 | 3.96 |
| N strip without tape | 0.924 | 0.944 | 0.141% | $424.4 \pm 4.7$ | 478 | 4.92 |

As shown in Table 2, thresh level increases the surface Vickers harness without peening. In order to investigate the tendency between the surface Vickers hardness and the threshold level, Figure 8 illustrates the relation between the surface Vickers harness without peening and the threshold level. It is known that Vickers hardness corresponds well to the yield stress of the material. Therefore, it is reasonable that the threshold level that deforms the material increases with Vickers hardness. As the number in the dataset was four and the correlation coefficient was 0.962, the probability of a non-correlation was 3.8% (Figure 8). When the probability of a non-correlation is less than 5%, it can be concluded that the relationship is significant. Thus, it can be concluded that the relationship between the surface Vickers harness without peening and the threshold level was significant. When an N strip with black aluminum tape was used, a larger LC was produced by a larger LA

due to the black aluminum tape, as the tape not only attenuated the impact but also increased LA.

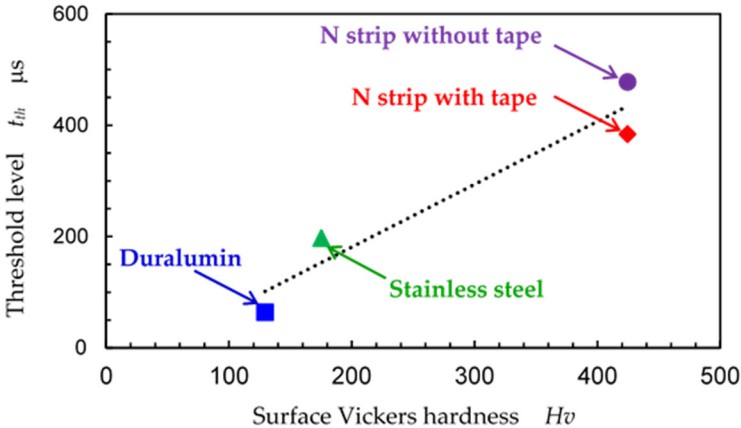

**Figure 8.** Relationship between the surface Vickers hardness and the threshold level.

## 3. Improvement of the Fatigue Strength of Welded Part by Laser Cavitation Peening

### 3.1. Experimental Apparatus and Procedures for the Fatigue Test of Welds

The welded specimen was treated by laser cavitation peening to demonstrate the improvement in the fatigue strength of welded metals by the proposed laser cavitation peening, as shown in Figure 9, which also illustrates the coordinates for residual stress measurement (blue arrows). The stainless-steel plate (Japanese Industrial Standards JIS SUS316L), with a U-notch and a radius of 1.5 mm, was machined and welded using single-pass Tungsten Inert Gas TIG welding. The plate was 95 × 180 mm and had a thickness of 3 mm. The five specimens for the plane-bending fatigue test were machined from one plate.

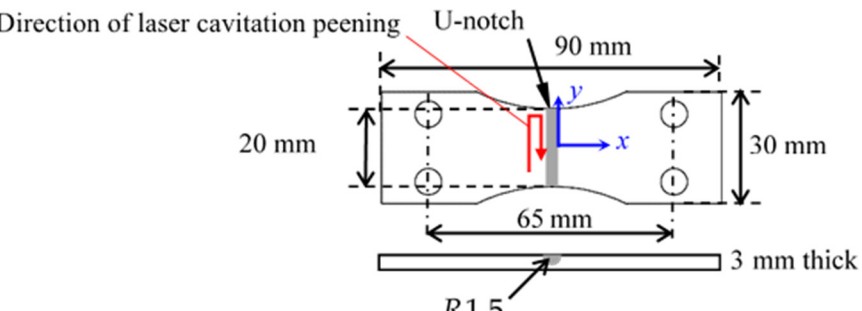

**Figure 9.** Geometry of the welded specimen for displacement-controlled plane-bending fatigue test, coordinates for residual stress measurement, and the direction of laser cavitation peening.

The specimen was treated by the proposed laser cavitation peening system, as shown in Figure 1. The wavelength of the Nd:YAG laser was 1064 nm, considering the above-mentioned results. The welded side was treated with a $d_L$ = 4 pulses/mm$^2$ using stepping motors, considering the previous report [11]. As the repetition frequency of the laser was 10 Hz, the specimen was treated widthwise at vs. = 5 mm/s and moved in $d_S$ = 0.5 mm steps in the longitudinal direction of the specimen, as shown in Figure 9 (red arrow).

The fatigue properties were examined using a conventional Schenk-type displacement-controlled plane-bending fatigue tester at $R = -1$. During plane-bending fatigue testing, displacement was produced by an eccentric wheel, and the applied stress was calculated from the bending moment, measured by a load cell [44]. The span length at a fixed point was 65 mm, and the test frequency was 12 Hz (Figure 9). The applied stress ($\sigma_a$) during the test was calculated from the bending moment ($M$), the width of the specimen ($b$) (i.e.,

20 mm), and the thickness ($\delta$) measured by a digital caliper, with an accuracy of 0.01 mm, as shown in Equation (5).

$$\sigma_a = \frac{6\,M}{b\,\delta} \tag{5}$$

As one of the major fatigue property factors in welded parts is residual stress, the residual stress near the welding line was evaluated using 2D X-ray diffraction with a two-dimensional detector [45]. The optimal conditions for residual stress measurement using 2D X-ray diffraction with specimen oscillation, considering the error, have been reported previously [46]. The X-ray tube used was Cr K$\alpha$ and was operated at 35 kV and 40 mA. The diameter of the collimator was 0.146 mm. The used lattice plane, ($h\,k\,l$), was the $\gamma$-Fe (2 2 0) plane and the diffraction angle without strain was 128 degrees. The diffraction from the surface was obtained by oscillating the specimen in the $\omega$-direction by 8 degrees during the measurement. The initial $\omega$ angle was 110 degrees. The 24 diffraction rings from the specimen at various angles were detected, and the exposure time per frame at each position was 20 min. It was reported that the residual stress at the surface treated by the submerged laser peening revealed the tension, and the residual stress at 30 $\mu$m below the surface well corresponded to the fatigue properties of the stainless steel [11]. Thus, the residual stresses at $z = 0$ $\mu$m and 30 $\mu$m were measured using electrochemical polishing.

*3.2. Results of the Welded Part Treated by Laser Cavitation Peening*

Figure 10 shows the welded part of the specimen with and without laser cavitation peening after the fatigue test. In both cases, the specimen fractured at the HAZ. The color of the surface treated by the laser cavitation peening became black due to oxidation caused by LA, and each spot deformed by LA and LC was revealed (Figure 10b). The deformation of the welding line was difficult to observe. The improvement in the fatigue properties could have been caused by introducing residual stress into the HAZ and reducing welding defects.

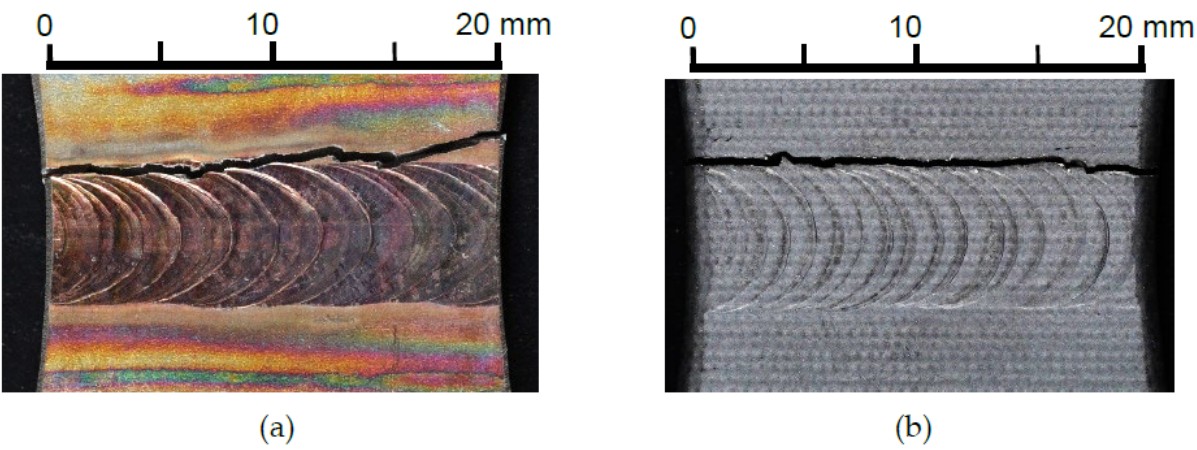

(a)  (b)

**Figure 10.** Fractured surface of a welded part. (**a**) Non-peened specimen ($\sigma_a$ = 229 MPa, $N$ = 688,800); (**b**) specimen treated by laser cavitation peening ($\sigma_a$ = 319 MPa, $N$ = 516,300).

The residual stress $\sigma_R$ as a function of the distance from the welding line $x$ at $z = 0$ and 30 $\mu$m in the $x$- and $y$-directions were calculated to investigate the effect of laser cavitation peening on the residual stress in the HAZ (Figure 11). The measurement location was mid-width, i.e., $y = 0$ (Figure 9). The tensile residual stress was introduced by laser ablation at $z = 0$ $\mu$m, and compressive residual stress was introduced by laser cavitation peening at $z = 30$ $\mu$m (Figure 11). It can be concluded that the tensile residual stress was introduced by laser ablation, but the compressive residual stress was introduced under the surface by the proposed laser cavitation peening. The distribution of the residual stress treated by laser cavitation peening changing with depth, compared with cavitation peening and shot peening, was revealed in [47].

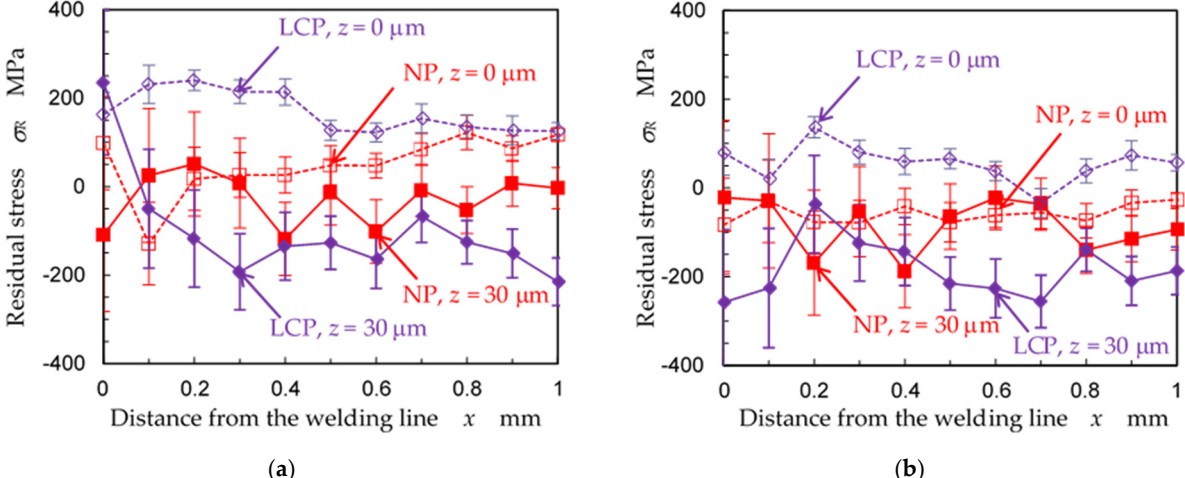

**Figure 11.** Residual stress at the surface $z = 0$ μm and $z = 30$ μm of non-peened (NP) and laser cavitation-peened (LCP) specimens. (**a**) $x$-direction; (**b**) $y$-direction.

*S-N* curves were obtained from the plane-bending fatigue test for non-peened (NP) and laser cavitation-peened (LCP) specimens to investigate the improvement in the fatigue properties of welded stainless steel with laser cavitation peening (Figure 12). The results of shot peening (SP), water jet peening (WJP), and cavitating jet peening (CJP), i.e., cavitation peening using cavitating jet, from reference [10] are also shown. The *S-N* of LCP was located on the upper right-hand side of NP (Figure 12); therefore, the fatigue life and strength were improved by LCP. When the fatigue strength was calculated using Little's method [48], the fatigue strength was 159 ± 20 MPa for NP and 244 ± 12 MPa for LCP. Namely, LCP improved the fatigue strength of welded stainless steel by 53%.

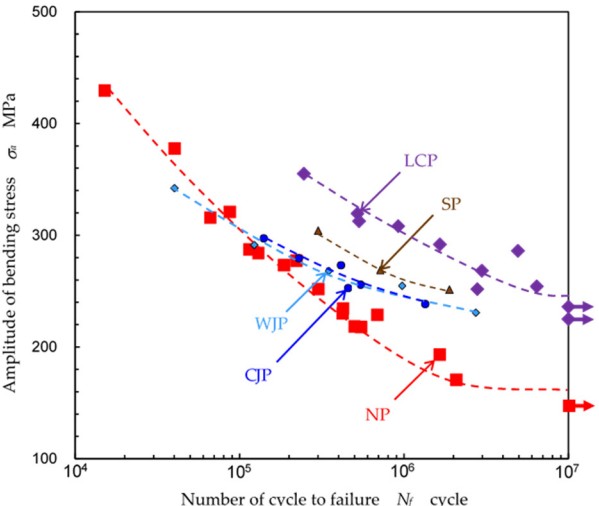

**Figure 12.** *S-N* diagram of welded specimens with and without laser cavitation peening compared with other peening methods. NP, non-peened; LCP, laser cavitation peening; SP, shot peening; WJP, water jet peening; CJP, cavitating jet peening (cavitation peening using cavitating jet). The SP, WJP, and CJP data were taken from [10].

Regarding the welding line observations and the residual stress results, the improvement in the fatigue strength mechanism was not from the decrease in stress concentration due to the deformation of the welding line, but from the introduction of compressive residual stress from LCP. The fatigue properties of LCP were better than the other peening methods, i.e., SP, WCP, and CJP (Figure 12).

Figure 13 shows the fractured surface of (a) a non-peened specimen and (b) a specimen treated by laser cavitation peening; in both cases, a crack was initiated at the void or a defect of welding near the surface, as the maximum tensile stress was applied at the surface during the bending fatigue test. For the non-peened specimen, the crack initiation point exhibited a wavy pattern at each weld, which would be a weak region; such defects are easily included during welding.

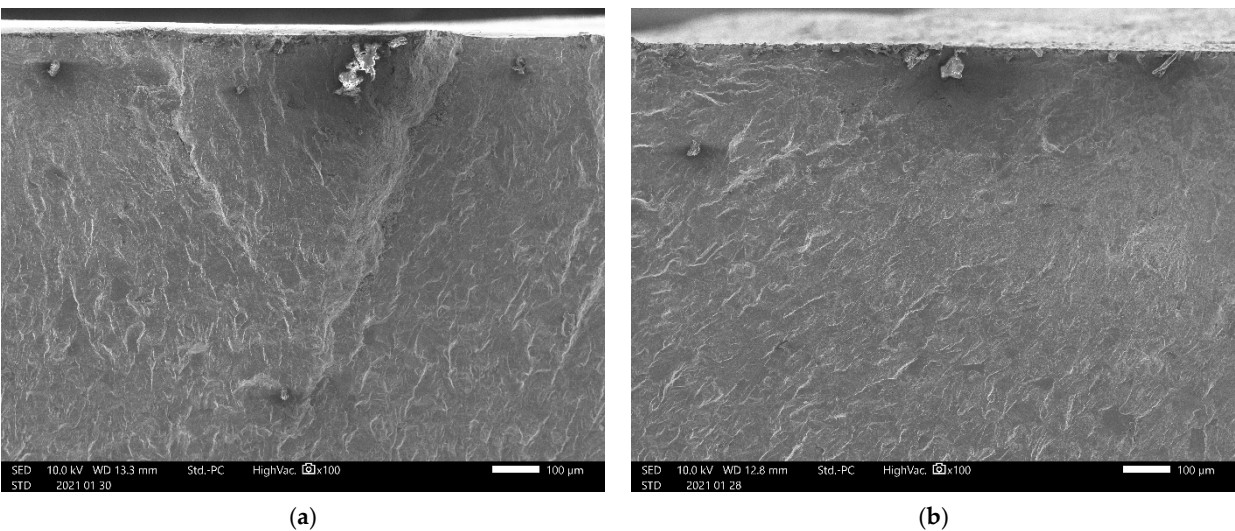

(a)                                                                                   (b)

**Figure 13.** Aspect of the fractured surface observed by SEM. (**a**) Non-peened specimen ($\sigma_a$ = 229 MPa, $N$ = 688,800); (**b**) specimen treated by laser cavitation peening ($\sigma_a$ = 319 MPa, $N$ = 516,300).

## 4. Conclusions

In order to investigate the effect of laser cavitation LC, which was generated after laser ablation LA, on submerged laser peening, impact and pressure wave were measured by using the handmade PVDF sensor and the conventional submerged shock wave sensor. The peening effect of the wave length of the Q-switched Nd:YAG laser was also investigated by comparing with the fundamental harmonic wave length, i.e., 1064 nm, which is normally used to generate cavitation, and the second harmonic wave length, i.e., 532 nm, which is normally used in conventional submerged laser peening to avoid absorption by water. After investigating LC, an improvement in the stainless-steel weld fatigue properties was demonstrated with the now named laser cavitation peening LCP. The results obtained can be summarized as follows:

(1) In order to investigate the peening effect of LA and LC, the impact measurement using the PVDF sensor was found to be better than that of the conventional submerged shockwave sensor, as the PVDF sensor could detect the second collapse of LC and the shockwave sensor could not. Both the LA and LC signals from the PVDF sensor were proportional to the arc height, and the correlation with the signal from the PVDF sensor was larger than that of the submerged shockwave sensor.

(2) The impact of LC measured by the PVDF sensor was larger than the impact of LC, although the pressure wave of LA measured by the submerged shockwave sensor was larger than that of LC. Namely, during optimized submerged laser peening, the impact induced by LC was more effective than that of LA, called laser cavitation peening (LCP).

(3) LCP, using a wavelength of 1064 nm, was more efficient than using 532 nm, as 40% of the pulse energy at 1064 nm was lost during wavelength conversion to 532 nm.

(4) The peening intensity was proportional to the volume of LC during LCP, considering the target metals' threshold level. The threshold level was roughly proportional to the surface Vickers hardness of the target metals without LCP.

(5) LCP could improve the fatigue properties of JIS SUS316L stainless steel welds by 53%.

**Funding:** This research was partly supported by the JSPS KAKENHI, grant numbers 18KK0103 and 20H02021.

**Institutional Review Board Statement:** Not applicable.

**Informed Consent Statement:** Not applicable.

**Data Availability Statement:** The data presented in this study are available on request from the author.

**Conflicts of Interest:** The author declares no conflict of interest.

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
