# Peer review of "Laser Cavitation Peening and Its Application for Improving the Fatigue Strength of Welded Parts"

_metals, doi:10.3390/met11040531_

Round 1

Reviewer 1 Report

The author proposes a very interesting investigation of the laser cavitation peening technique for the improvement of fatigue performances of welding, related to the compressive residual stress field generated by the peening itself. The results are well presented, but few issues need to be clarified and some other addressed carefully.

The main concern is about the cavitation itself that, as stated in literature, is a negative effect in the laser peening process. The cavitation bubbles usually create vibrations on the tested coupons, especially in thin-walled components as the tested welded coupons, and scattering of the laser beam, thus reducing the effectiveness of the peening process because of the dispersion of the induced stresses. This is not clear in the paper and how this problem is overcome.

About the introduction, the literature review is well defined, but the motivations are not clear. Some sentences must be clarified, as in line 30 “It is worthwhile a new peening technique….”. Many peening techniques are widely available but a comparison with the laser cavitation peening subject of this paper is missing, so are the advantages or improvement.

Furthermore, there is a lot of confusion in the description of the laser peening terminology of the laser cavitation peening, laser peening etc. The discussion from line 54 to 83 should be reviewed.

In the par.2, how the PVDF sensor has been calibrated? The description of the experiment lacks some details, such as the thickness of the coupons. How many coupons have been tested for each material or configuration?

Why 4 pulses per square mm? Is there a full overlap for every pulse sequence?

Equations are not numbered and difficult to read, please check their alignment and font.

In line 142, while it is clear from picture 2b the second collapse for the lambda 532 nm, the same condition for lambda 1,064, defined at t=1,52 msec, is not confirmed in figure 3b because of the higher noise in the signal. The overall discussion should be improved because too confused.

Line 160, please clarify the definition of maximum diameter of LC.

Figure 4 contains many values not described in the text. Line 163: where is the laser’s focus in the water? Which kind of water has been used? Which is the meaning of the different colours in the graph? Why are there two different colours for Water 10°C? etc..etc…

Line 171, the figure 5 doesn’t show the max diameter but the time, therefore is not correct the caption, or the diameter should be calculated

Figure 7 (a) and (b) are too similar for different values of n, are them correct?

Par 3: how many coupons have been tested? From figure 12 it seems that a single coupon has been tested for a fixed amplitude value. The most important comparison, with standard laser shock peening, is missing.

How is the through-the thickness stress distribution after the peening process? Why laser ablation creates tensile stresses?

Many errors in English language and many sentences need a deep review

Reviewer 2 Report

The submission requires minor language corrections. For example "...the pulse laser was reflected mirrors..", "...threshold level was existed...", "thresh level...", "...stepping motors..."

Author Response

Comments and Suggestions for Authors: The submission requires minor language corrections. For example "...the pulse laser was reflected mirrors..", "...threshold level was existed...", "thresh level...", "...stepping motors..."

 Response:  My manuscript was proofread and corrected by MDPI’s English editing service.

Reviewer 3 Report

Line 26: The fatigue strength of HAZ is higher than the one of base metal or lower? If it would be higher, any surface treatment would be required.

Line 124: Give the geometry and dimensions of the treated samples.

Line 160: What is the developing time of LC? Is it the time needed to attain the maximum diameter of the bubble?

Line 187: Why have you used an N strip with tape? Is there any sense to use a tape on it? What is a black aluminum tape? Clarify these points.

Fig. 7b) h in proportional to (tD-th)3. You have indicated h proportional to tD3.

Line 218: In the case of the N strip with tape, shouldn´t you consider the hardness of the surface tape?

Line 250: Give the filler metal you have used and also the welding parameters. Have you performed the weld in a single pass? I guess you have not performed a real weld, but you have filled the machined U notch instead, haven´t you.

Line 290: Deformation of the welding line is difficult to be observed in this figure. Anyway, deformation impacts are visible in Fig. 10 b), so I guess laser cavitation may also probably reduce welding defects. Don´t you think so?

Figure 11: Where have you performed these meassurements? At y=0 location, mid-width? (Fig. 9)

Round 2

Reviewer 1 Report

The paper has been improved consistently with the comments from the reviewer, therefore is worth of publishing